# Geostatistical analysis to guide treatment decisions for soil-transmitted helminthiasis control in Uganda

Bryan O. Nyawanda[1,2], Kristin M. Sullivan[3], Benjamin Tinkitina[4], Prudence Beinamaryo[4], Betty Nabatte[4], Hilda Kyarisiima[4], Alfred Mubangizi[4], Paul M. Emerson[3], Jürg Utzinger[1,2], Penelope Vounatsou[1,2]*

1 Swiss Tropical and Public Health Institute, Allschwil, Switzerland, 2 University of Basel, Basel, Switzerland, 3 Children Without Worms, The Task Force for Global Health, Decatur, Georgia, United States of America, 4 Vector Borne and Neglected Tropical Diseases Division, Ministry of Health, Kampala, Uganda

* penelope.vounatsou@unibas.ch

## Abstract

### Background

Soil-transmitted helminth (STH) infections remain a public health problem in Uganda despite biannual national deworming campaigns implemented since the early 2000s. Recent surveys have indicated a heterogeneous STH infection prevalence, suggesting that the current blanket deworming strategy may no longer be cost-effective. This study identified infection predictors, estimated the geographic distribution of STH infection prevalence by species, and calculated deworming needs for school-age children (SAC).

### Methodology

Bayesian geostatistical models were applied to STH survey data (2021–2023) for each species (i.e., *Ascaris lumbricoides*, hookworm, and *Trichuris trichiura*). Climatic, environmental, and socioeconomic predictors were obtained from remote sensing sources, model-based databases, and demographic and health surveys. Prevalence was predicted on a 1 × 1 km² grid across Uganda, and district-level estimates were used to classify each district into treatment frequency categories and to determine its deworming tablet requirements.

### Principal findings

The national prevalence of *A. lumbricoides*, *T. trichiura,* and hookworm was estimated at 5.0% (95% Bayesian credible interval [BCI]: 0.8–11.8%), 3.5% (0.7–9.3%), and 7.2% (5.7–11.1%), respectively. The overall prevalence of any STH infection was 14.3% (9.6–21.8%). High intra-district variation in prevalence was observed. Of 146

which permits unrestricted use, distribution, and reproduction in any medium, provided the original author and source are credited.

**Data availability statement:** The survey data used in this paper are owned by the government of Uganda. Researchers may request access to de-identified datasets from the Vector Borne and Neglected Tropical Diseases Division at the Ministry of Health (vcdmoh@gmail.com). Due to legal restrictions, the dataset cannot be made publicly available. These restrictions are imposed by the Ministry of Health in accordance with national data protection and ethics guidelines. However, we provide references and links within the manuscript to other publicly accessible datasets that may be useful for related research.

**Funding:** This research was funded by sponsorship provided to Children Without Worms (Decatur, Georgia, United States of America) by Johnson & Johnson (New Brunswick, New Jersey, United States of America) to support the STH control efforts of the Vector Borne and Neglected Tropical Disease Division, Ministry of Health, Uganda. The funders played no role in the study design, data collection and analysis, publication decision, or manuscript preparation.

**Competing interests:** I have read the journal's policy and the authors of this manuscript have the following competing interests: Kristin M. Sullivan receives a salary from The Task Force for Global Health, which receives funding from GSK and Johnson & Johnson, the manufacturers of albendazole and mebendazole, respectively. The findings and conclusions in this article are those of the authors and do not necessarily reflect the official position of the funders or the institutions they represent. The other authors have declared that no competing interests exist.

implementation units (136 districts and 10 cities), 49 require twice-year treatment, 34 once-yearly treatment, 61 every other year treatment, and 2 had a prevalence <2%, indicating treatment suspension or event-based treatment. Approximately 17 million tablets will be needed for preventive chemotherapy aimed at SAC in 2025.

## Conclusions/significance

The prevalence of STH infection has declined considerably across Uganda compared to the early 2000s. However, deworming needs remain heterogeneous across districts. Through geostatistical modeling, districts were classified according to the latest World Health Organization's (WHO) treatment guidelines. This approach optimizes treatment distribution and allows for prioritization of populations with the greatest needs. We estimated that tablet requirements are approximately 40% lower compared to the current twice-a-year deworming regimen, which contributes towards WHO's goal of halving the number of tablets required for preventive chemotherapy by 2030.

## Author summary

Morbidity due to parasitic worm infections (roundworm, whipworm and hookworm) remains a public health concern in Uganda despite biannual deworming campaigns. Results from recent surveys indicate that infection prevalence varies across the country, suggesting that blanket treatment may no longer be the most efficient use of resources. This study used Bayesian geostatistical models to estimate the prevalence of infection cause by the above parasitic worms, based on survey data collected from 2021 to 2023. The analysis identified key predictors, including climatic, environmental, and socioeconomic factors, and classified districts according to their treatment frequency category. Population-adjusted national prevalence was estimated at 5.0% for roundworm, 3.5% for whipworm, and 7.2% for hookworm, with 49 out of 146 districts exceeding 20% prevalence. An estimated 17 million deworming tablets will be needed for preventive chemotherapy distributions in 2025. These findings support targeted treatment strategies to improve efficiency and prioritize high-risk areas, aligning with World Health Organization guidelines.

## Introduction

Soil-transmitted helminth (STH) infections are the most widespread neglected tropical diseases (NTDs) with roundworms (*Ascaris lumbricoides*), whipworms (*Trichuris trichiura*), and hookworms (*Ancylostoma duodenale* and *Necator americanus*) affecting hundreds of millions of people, particularly in tropical and subtropical regions with poor sanitation [1–3]. Globally, an estimated 24% of the population (approximately 1.5 billion people) are infected, with the highest burden in low- and middle-income

countries [3,4]. Sub-Saharan Africa accounts for 39% of global cases, making it one of the most heavily affected regions [5]. STH infections are particularly prevalent in children, causing intestinal bleeding, malnutrition, anemia, stunting, and cognitive impairment [6–8].

In 2001, the World Health Organization (WHO) included STH in the NTD control program, recommending preventive chemotherapy (PC) as a primary intervention [9]. PC is typically provided to school-age children (SAC; aged 5–14 years) through periodic school-based distributions using a single tablet of albendazole (400 mg) or mebendazole (500 mg). While PC, improved sanitation, and social and economic development have reduced the global burden of STH, reinfection due to poor sanitation remains a major challenge [10,11]. WHO emphasizes reassessment of STH prevalence following 5 years of high-coverage deworming to inform targeted interventions and optimize PC frequency, aiming to reduce tablet needs and establish efficient control programs for preschool-age children (pre-SAC; aged 1–4 years) and SAC [11–13]. As part of this strategy, WHO recommends the use of geostatistical modeling to guide spatial targeting of interventions [14].

Compared to the high (>50%) STH infection prevalence reported among SAC in Uganda from 1998 to 2005, recent surveys in 10 districts have revealed a decline in all but one district, where the prevalence remained high (>60%) [15,16]. Uganda continues to offer universal, twice-yearly PC to SAC [16]. However, this may no longer be necessary in light of these recent findings. To maximize the insights gained from these and earlier survey data, and to target interventions to those most in need, predictive modeling can serve as a valuable complementary tool for estimating prevalence in areas that lack survey data and improving our understanding of infection distribution. Bayesian geostatistical modeling (BGM) leverages climatic, environmental, socioeconomic, soil, and water, sanitation, and hygiene (WASH) data to predict STH prevalence [7,17–19].

Using survey data from the Ugandan NTD control program, we employed BGM to (i) identify predictors associated with infection prevalence of STH; (ii) predict species-specific prevalence on a $1 \times 1$ km$^2$ gridded surface across Uganda; and (iii) classify districts by their WHO-recommended treatment frequency in order to estimate the tablets required for PC distribution to SAC. Our results provide actionable insights to optimize treatment strategies and enhance STH control efforts in Uganda, and may guide other countries to adopt a similar approach.

## Methods

### Ethics statement

The protocols for the surveys were reviewed and approved by the Vector Control Division Research Ethics Committee and the Uganda National Council of Science and Technology, as previously described [15,16]. Survey participation was voluntary. Participants provided assent, while their parents or guardians provided verbal and written informed consent. For this secondary data analysis, ethical approval was deemed unnecessary by Uganda's Vector Control Division and the Swiss Tropical and Public Health Institute.

### Country profile

Uganda is a landlocked country covering 241,038 km$^2$ with an estimated population of 46 million people in 2024 [20]. It shares borders with Kenya (East), Tanzania and Rwanda (South), the Democratic Republic of the Congo (West), and South Sudan (North). The country is comprised of 136 districts grouped into 4 regions, and has 10 cities (making a total of 146 implementation units). Its diverse ecosystem ranges from tropical rainforests in the South to savannah and semi-desert landscapes in the North. Altitudes span 620–5,094 m above the mean sea level, with mean annual temperatures between 14°C and 32°C. Uganda experiences two rainfall seasons: short rains (March–May) and long rains (September–December).

### Data sources

The STH data for this analysis were obtained from three survey rounds: school-based surveys in 750 locations across 38 districts in 2021, community-based surveys in 150 locations across 5 districts in March–May 2022 [16], and school-based

surveys in 25 locations in 5 districts in November 2023 [15]. Detailed methodologies for the 2022 and 2023 survey rounds have been described elsewhere [15,16]. The current analysis focuses on SAC.

STH data were combined with climatic, environmental, soil, WASH, and treatment data from the respective survey locations. Climate variables, including land surface temperature (LST) and rainfall, were obtained from Moderate-resolution Imaging (MODIS) [21] and the Climate Hazards Group InfraRed Precipitation with Station (CHIRPS) database [22], processed as monthly averages, and used to derive 19 bioclimatic variables (Table A in S1 Appendix). Vegetation index (normalized difference vegetation index [NDVI] and enhanced vegetation index [EVI]) were extracted from MODIS at 1 × 1 km$^2$ spatial resolution [23]. Nightlight data, a proxy for urbanization and socioeconomic status (SES), were obtained from the National Oceanic and Atmospheric Administration – Visible Infrared Imaging Radiometer Suite (NOAA–VIIRS) [24].

WASH indicators as defined by the WHO/UNICEF (United Nations Children's Fund) Joint Monitoring Programme (JMP) – i.e., proportion of households with improved sanitation, with access to an improved drinking water source, and practicing open defecation – were based on the 2016 Demographic and Health Surveys (DHS) [25], the 2018–2019 Malaria Indicator Survey (MIS) [26], and 2016–2021 trachoma prevalence surveys supported by Tropical Data [27]. We used cluster-level data from these surveys in a BGM to generate prevalence maps of the WASH indicators (Fig A in S2 Appendix). Treatment coverage data for STH were downloaded from WHO's Expanded Special Project for the Elimination of Neglected Tropical Diseases (ESPEN) [5] and used to estimate the average proportion of individuals treated between 2020 and 2022 per district. Lymphatic filariasis (LF) treatment was considered but not included as a covariate, as the last LF treatment round was administered in six districts in 2018 [5]. The time lapse between this treatment together with its partial implementation was considered to have negligible influence on STH prevalence. Elevation data were obtained from the Shuttle Radar Topographic Mission (SRTM) [28], and soil composition data from the International Soil Reference and Information Centre (ISRIC) [29]. All predictors were extracted and processed at each survey location. The expected SAC population in 2025 was sourced from the Ugandan Ministry of Health. Data sources and their spatio-temporal resolutions are summarized in Table A in S1 Appendix.

## Statistical analysis

Covariates with a variance inflation factor (VIF) >10 were excluded to address collinearity [30]. Continuous covariates were standardized (scaled by mean and standard deviation [SD]) to improve model stability. Initially, species-specific bivariate Bayesian geostatistical binomial and zero-inflated binomial (ZIB) regression models were fitted incorporating spatial and – optionally – non-spatial locational random effects to: (i) identify the best model formulation (binomial or ZIB with or without the presence of non-spatial random effects) and (ii) select an initial set of predictors that optimized goodness of fit and in-sample predictive performance, evaluated using the Watanabe-Akaike information criterion (WAIC) [31]. Subsequently, for each STH species, the models arising from all combinations of those initial predictors were fitted (i.e., 32,752 models). The top models with similar performance (WAIC values within 4 units of the lowest WAIC, i.e., ΔWAIC ≤4) were selected for (i) predictor analysis and (ii) identification of the most parsimonious predictive model.

The predictor analysis involved calculating how frequently each predictor was included in the top models with a statistically important positive effect (i.e., 95% Bayesian credible interval [BCI] for the coefficient entirely above zero), a statistically important negative effect (i.e., 95% BCI entirely below zero), or no statistically important effect (i.e., 95% BCI including zero).

The most parsimonious predictive model was selected based on out-of-sample predictive performance. To evaluate this, the dataset was split: 85% of survey locations formed the training set to fit the model, while the remaining 15% served as the test data. This split maximized available training data given the sparsity of survey data. Predictive performance was assessed using root mean square error (RMSE) and mean absolute error (MAE), and the model yielding the lowest RMSE and MAE was chosen for final predictions. This model was employed in Bayesian kriging to predict prevalence across a 1 × 1 km$^2$ grid covering Uganda. The median of the posterior predictive distribution at each grid cell

generated the species-specific prevalence maps, while the 25th and 75th percentiles produced the interquartile range for the species-specific uncertainty maps. Details on the model specifications are provided in S3 Appendix.

Assuming independence among STH species, we calculated the prevalence of any STH species ($p_s$) at the grid-cell level using the formula $p_s = p_A + p_T + p_h - p_A \times p_T - p_A \times p_h - p_T \times p_h + p_A \times p_T \times p_h$, where $p_A$, $p_T$, and $p_h$ indicate the predicted prevalence of *A. lumbricoides*, *T. trichiura*, and hookworm infections, respectively [32]. Because this assumption overestimates the combined STH prevalence by 0.6% for every 10% increase in prevalence [33], we applied a correction factor: $p_{ATh} = p_s \div 1.06$. To classify any STH prevalence into endemicity levels, we generated five probability maps corresponding to the following categories: <2%, 2–9.9%, 10–19.9%, 20–49.9%, and ≥50% [34]. We drew 200 samples from the predictive posterior distribution for each grid cell, assigned a value of 1 if the sample fell within a given category or zero otherwise [35], and used the frequency of 1s to estimate the probability of each grid cell belonging to that endemicity category [7].

WHO classifies districts (formally called "implementation units") into different treatment frequency categories based on the prevalence of any STH infection among SAC [34]. According to WHO guidelines, the recommended treatment frequency is as follows: no or event-based treatment if the prevalence is <2%; treatment once every two years for prevalence between 2% and 9.9%; treatment once per year for prevalence between 10% and 19.9%; and treatment twice per year for prevalence ≥20% [14]. Because the option of event-based treatment in districts with a prevalence of <2% is a recent WHO recommendation and has not yet been implemented by countries, for this analysis, we assumed that districts with a prevalence of <2% would suspend treatment, and thus no tablets were needed in these districts.

To assign each district to one of these treatment categories, we calculated the probability of the district belonging to the corresponding endemicity categories (i.e., <2%, 2–9.9%, 10–19.9%, and ≥20%). Specifically, the 200 samples were aggregated to obtain the district-level prevalence by averaging the values across the grid cells within the district. Each sample was then classified into one of the four treatment categories. The frequency distribution of the sample classifications was used to estimate the probabilities of a district belonging to each treatment category. A district was assigned to the treatment class with the highest probability, provided that the difference between this probability and the second-highest probability exceeded 12.5 percentage points. Given that every district had a 25% chance of falling in any of the four categories *a priori*, a group with a difference of more than 12.5 percentage points (i.e., 50% of 25%) offered greater certainty that the district belonged to that category. If the difference was less than 12.5 percentage points, the district was conservatively assigned to the next treatment category with the highest endemicity level according to the classification order. The annualized tablet needs for 2025 at the district level were estimated by multiplying the SAC population in each district by the required number of PC rounds corresponding to their treatment category (i.e., 0, 1/2, 1, or 2 for <2%, 2–9.9%, 10–19.9%, and ≥20% endemicity levels, respectively). National prevalence estimates were derived by averaging the predicted values across all grid cells for each of the 200 samples, resulting in 200 national-level estimates. These were then used to calculate the median prevalence and the corresponding BCI from the 2.5th and 97.5th percentiles.

All statistical analyses were conducted using R software version 4.3.2 [36]. Models were fitted, and Bayesian kriging was conducted using integrated nested Laplace approximation (INLA) [37]. Samples were drawn from the posterior predictive distributions of prevalence using INLA's posterior sampling function. We refer to statistically significant results as statistically important, since the term "significance" is not appropriate in the context of Bayesian inference.

## Results

### Descriptive analyses

Survey data from SAC were available in 966 locations (schools or households) in 56/136 (41.2%) districts. Notably, the northeastern part of the country lacked survey data. Of the sampled locations, 753 (78.0%), 826 (85.5%), and 380 (39.3%) reported zero cases of *A. lumbricoides*, *T. trichiura*, and hookworm, respectively. The observed prevalence estimates for *A. lumbricoides*, *T. trichiura*, hookworm, and any STH were 2.0%, 1.2%, 7.2%, and 10.2% respectively.

## Variable and model selection

Annual mean temperature, temperature annual range, mean temperature of wettest quarter, mean temperature of driest quarter, mean temperature of warmest quarter, mean temperature of coldest quarter, annual precipitation, precipitation seasonality, EVI, and altitude were excluded from the analysis because they had VIF > 10.

Results from the bivariate binomial and ZIB models with and without the non-spatial random effect are summarized in Tables B, C and D, in S1 Appendix. Overall, the geostatistical models with non-spatial random effects had lower Deviance information criterion (DIC) and WAIC for all the species. For *A. lumbricoides* and *T. trichiura,* the ZIB models with predictors in the zero inflation component had lower DIC and WAIC compared to the binomial and ZIB without predictors in the zero inflation component.

The frequency with which each predictor was included in the top models with similar WAIC values is summarized in Table 1. An increase in minimum temperature in the coldest month (27/71 models) and sandy soil structure (39/71 models) were negatively associated with *A. lumbricoides* infection. In contrast, organic carbon stocks had a statistically important positive relationship with *A. lumbricoides* infection in 16/71 models. Mean diurnal range was negatively associated with *T. trichiura* infection in 2/14 models. In contrast, precipitation of the driest month, slope, and open defecation were positively associated with *T.trichiura* infection in 3/14, 6/14, and 3/14 models, respectively. Additionally, nightlights were statistically important and negatively associated with hookworm infection in 170/200 models. In contrast, cation exchange capacity and treatment were statistically important and positively associated with hookworm infection in 136/200 and 200/200 models, respectively. The geographic distribution of hookworm prevalence by treatment coverage is displayed in Fig B in S2 Appendix.

The best predictive model for *A. lumbricoides* included the minimum temperature of the coldest month, mean diurnal range for temperature, bulk density of the fine Earth fraction, the proportion of sand particles, the proportion of clay particles, slope, the proportion of households with improved sanitation, and the proportion treated using albendazole or mebendazole (RMSE = 0.131 and MAE = 0.061). For *T. trichiura,* the best model included the proportion of sand particles, bulk density of the fine Earth fraction, precipitation of the wettest quarter, precipitation of the driest month, soil organic carbon content, cation exchange capacity of the soil, mean diurnal range for temperature, and treatment using albendazole or mebendazole (RMSE = 0.088 and MAE = 0.029). The best model for hookworm included nightlights, cation exchange capacity, proportion of clay particles, and treatment using albendazole or mebendazole (RMSE = 0.117 and MAE = 0.068). The top five best predictive models for each species are summarized in Table 2, and the parameter estimates for the champion models by species are summarized in Table E in S1 Appendix. For all models, the posterior distribution of the parameters differed substantially from their priors, suggesting that the estimates were data-driven and not influenced by the prior assumptions.

## Geographic distribution

Fig 1 displays the geographic distribution of STH species-specific observed prevalence (left), predicted prevalence (center), and uncertainty (right). For *A. lumbricoides*, most districts had a predicted prevalence of <2%. However, districts West of Lake Victoria, including Kyotera, Masaka, Bukomansimbi, and Kalungu, had prevalences ≥20%, while the prevalence in Kisoro, Rubanda, and parts of Kasese districts was estimated at ≥50%. The predicted prevalence of *T. trichiura* was < 2% in most districts, except for Kisoro and Kalangala, where it exceeded 20%. Similarly, most districts had a predicted hookworm prevalence of <10%. Districts with predicted hookworm prevalence ≥20% tended to be South of Lake Kyoga and North of Lake Victoria (Kamuli, parts of Luuka, Kaliro, Buyande, Kayunga, and Manafwa). For all species, the uncertainty interquartile range (IQR) was wider in areas where surveyed locations showed high variation in prevalence estimates.

Fig 2 shows the predicted prevalence of infection with any STH species and the pixel-level endemicity category. Districts in the northwestern part of Uganda had a higher probability of being in the < 2% category. Many parts of the country

**Table 1. Predictor analysis based on the frequencies of the predictors in the top similar models as defined by the Watanabe-Akaike information criterion (WAIC).**

| Predictor | *Ascaris lumbricoides* (71 models) | | | *Trichuris trichiura* (14 models) | | | Hookworm (200 models) | | |
|---|---|---|---|---|---|---|---|---|---|
| | Not statistically important | Important and negative | Important and positive | Not statistically important | Important and negative | Important and positive | Not statistically important | Important and negative | Important and positive |
| Climatic: mean diurnal range | 35 | 0 | 0 | 1 | 2 | 0 | 0 | 0 | 0 |
| Climatic: isothermality | 10 | 0 | 0 | 0 | 0 | 0 | 190 | 1 | 0 |
| Climatic: temperature seasonality | 14 | 0 | 0 | 4 | 0 | 0 | 0 | 0 | 0 |
| Climatic: min temperature of coldest month | 44 | 27 | 0 | 4 | 0 | 0 | 28 | 0 | 0 |
| Climatic: precipitation of wettest month | 0 | 0 | 0 | 2 | 0 | 0 | 59 | 0 | 0 |
| Climatic: precipitation of driest month | 0 | 0 | 0 | 11 | 0 | 3 | 0 | 0 | 0 |
| Climatic: precipitation of wettest quarter | 7 | 0 | 0 | 14 | 0 | 0 | 65 | 0 | 0 |
| Environmental: NDVI* | 0 | 0 | 0 | 0 | 0 | 0 | 48 | 0 | 0 |
| Environmental: night lights | 24 | 0 | 0 | 0 | 0 | 0 | 29 | 170 | 0 |
| Environmental: slope | 34 | 0 | 0 | 1 | 0 | 6 | 0 | 0 | 0 |
| Soil: bulk density of the fine Earth | 31 | 0 | 0 | 9 | 0 | 0 | 8 | 0 | 0 |
| Soil: cation exchange capacity | 0 | 0 | 0 | 13 | 0 | 0 | 30 | 0 | 136 |
| Soil: volumetric fraction of coarse fragments | 11 | 0 | 0 | 2 | 0 | 0 | 0 | 0 | 0 |
| Soil: soil organic carbon content | 0 | 0 | 0 | 3 | 0 | 0 | 3 | 0 | 0 |
| Soil: total nitrogen | 22 | 0 | 0 | 0 | 0 | 0 | 37 | 0 | 0 |
| Soil: organic carbon stocks | 27 | 0 | 16 | 0 | 0 | 0 | 77 | 0 | 0 |
| Soil: clay | 31 | 0 | 1 | 11 | 0 | 0 | 56 | 10 | 0 |
| Soil: sand | 8 | 39 | 0 | 14 | 0 | 0 | 25 | 0 | 0 |
| Soil: silt | 0 | 0 | 0 | 0 | 0 | 0 | 21 | 0 | 0 |
| WASH**: improved sanitation | 43 | 0 | 0 | 0 | 0 | 0 | 0 | 0 | 0 |
| WASH: open defecation | 24 | 0 | 0 | 4 | 0 | 3 | 129 | 0 | 0 |
| Treatment (albendazole or mebendazole) | 71 | 0 | 0 | 14 | 0 | 0 | 0 | 0 | 200 |

*NDVI, normalized difference vegetation index; **WASH, water, sanitation, and hygiene.

Note - the row totals for each species may not add up to the number of models because the models had different variable combinations and as such, some variables were excluded in some of the models.

**Table 2. Summary of validation metrics for the top five models for each soil-transmitted helminth species analyzed.**

| Model | RMSE | MAE | WAIC |
|---|---|---|---|
| *Ascaris lumbricoides* **(5/71 models)** | | | |
| BIO6 + imprSan + BIO2 + bdod + sand + slope + clay + proptrtd | 0.13117 | 0.06051 | 1144.493 |
| BIO6 + BIO2 + bdod + sand + clay + proptrtd | 0.13336 | 0.06313 | 1140.975 |
| BIO6 + imprSan + BIO2 + bdod + slope + proptrtd | 0.13658 | 0.06574 | 1140.921 |
| BIO6 + imprSan + BIO2 + bdod + sand + clay + lights + proptrtd | 0.13768 | 0.06611 | 1144.060 |
| BIO6 + imprSan + BIO2 + openDef + slope + clay + proptrtd | 0.14016 | 0.06778 | 1143.538 |
| *Trichuris trichiura* **(5/14 models)** | | | |
| sand + bdod + BIO16 + BIO14 + BIO4 + soc + cec + BIO2 + proptrtd | 0.08758 | 0.0291 | 828.597 |
| sand + bdod + openDef + BIO16 + slope + BIO14 + clay + cec + proptrtd | 0.08808 | 0.02974 | 820.699 |
| sand + bdod + openDef + BIO16 + BIO14 + BIO4 + clay + cec + proptrtd | 0.08916 | 0.03002 | 822.245 |
| sand + bdod + BIO6 + openDef + BIO16 + BIO14 + clay + cec + proptrtd | 0.08936 | 0.03103 | 823.405 |
| sand + openDef + BIO16 + slope + BIO14 + clay + proptrtd | 0.08970 | 0.03064 | 826.617 |
| Hookworm (5/200 models) | | | |
| lights + cec + clay + proptrtd | 0.11720 | 0.06797 | 2931.279 |
| lights + cec + BIO3 + soc + openDef + nitro + proptrtd | 0.11725 | 0.06830 | 2929.725 |
| lights + cec + BIO3 + openDef + nitro + proptrtd | 0.11744 | 0.06823 | 2928.484 |
| lights + cec + clay + BIO13 + proptrtd | 0.11747 | 0.06809 | 2931.891 |
| lights + cec + BIO3 + openDef + BIO16 + clay + BIO6 + proptrtd | 0.11750 | 0.06811 | 2929.968 |

RMSE, root mean square error; MAE, mean absolute error; WAIC, Watanabe-Akaike information criterion.

BIO2, mean diurnal range; BIO3, isothermality; BIO4, temperature seasonality;

BIO6, min temperature of coldest month; BIO14, precipitation of driest month;

BIO16, precipitation of wettest quarter; BDOD, bulk density of the fine Earth.

CEC, cation exchange capacity; soc, soil organic carbon content;

imprSan, improved sanitation; openDef, open defecation; proptrtd, treatment coverage.

fell into the 2–9.9% category, while Kisoro and Rubanda were in the very high (≥50%) category. There was significant variation in district-level prevalence of any STH infection in several districts, with drawn samples distributed across multiple treatment frequency categories, including Abim, Napak, Busia, Karenga, and Kitgum, among others (Table F in S1 Appendix). However, for most districts, the classification was clear, with most samples falling into a single category when the probability exceeded 50%. For example, out of the 200 samples drawn from the predictive posterior distribution for Maracha, Amuru, Kiruhura, and Kayunga districts, 74%, 71%, 90%, and 77% fell into the very low, low, moderate, and high/very high endemicity categories, respectively (Table F in S1 Appendix).

## Prevalence and tablet needs

The estimated national predicted prevalence of *A. lumbricoides*, *T. trichiura*, and hookworm was 5.0% (95% BCI: 0.8–11.8%), 3.5% (0. 7–9.3%), and 7.2% (5.7–11.1%), respectively, while the prevalence of any STH infection was estimated at 14.3% (9.6–21.8%). Of the 146 implementation units, 63 were categorized as <10% and 49 as ≥20% (Table 3 and Table F in S1 Appendix).

Uganda is projected to have a population of approximately 14.3 million SAC in 2025. Based on our model-based district-level endemicity classification, we estimated that approximately 17.0 million tablets will be required for PC in 2025 (Table 3). A map of districts by treatment rounds is shown in Fig 3, while the associated district-level predicted prevalence and tablet requirements are presented in Table F in S1 Appendix.

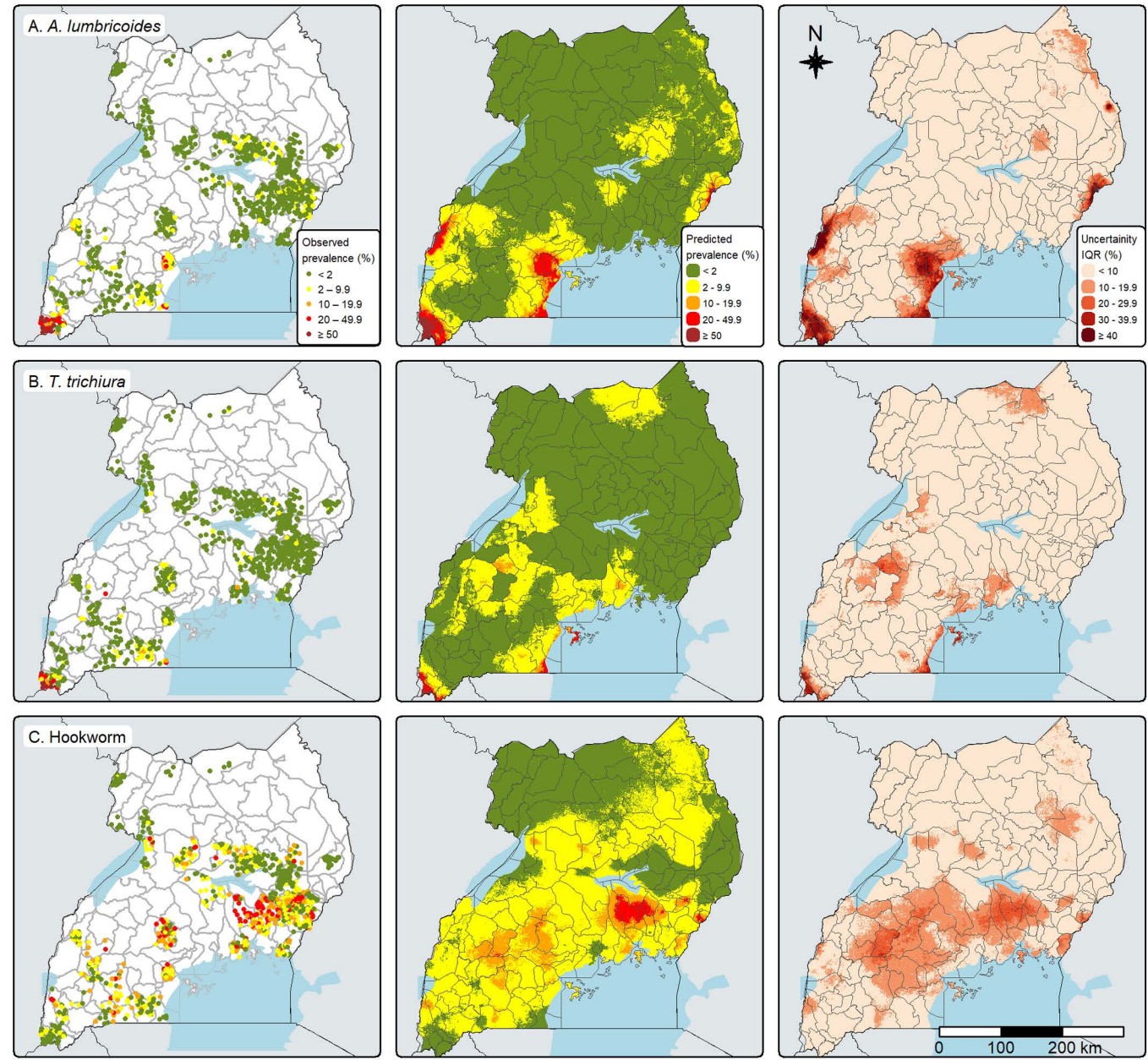

**Fig 1. Maps of the observed prevalence (left), predicted prevalence (median of the posterior predictive distribution; center), and prediction uncertainty (interquartile range; right) for the three soil-transmitted helminth species.** These maps were created using the tmap-package in R, and the basemap shapefiles were downloaded from ESPEN (2022). https://espen.afro.who.int/maps-data/data-query-tools/cartography-database [5] and https://data.humdata.org/dataset/cod-ab-uga [38].

## Discussion

Studies conducted in Uganda between 1998 and 2005, prior to large-scale deworming campaigns, found STH infection prevalence exceeding 50%. Our findings suggest that the national control program has been highly effective in reducing STH infection prevalence in the past two decades, with the national predicted prevalence estimated from our analysis at

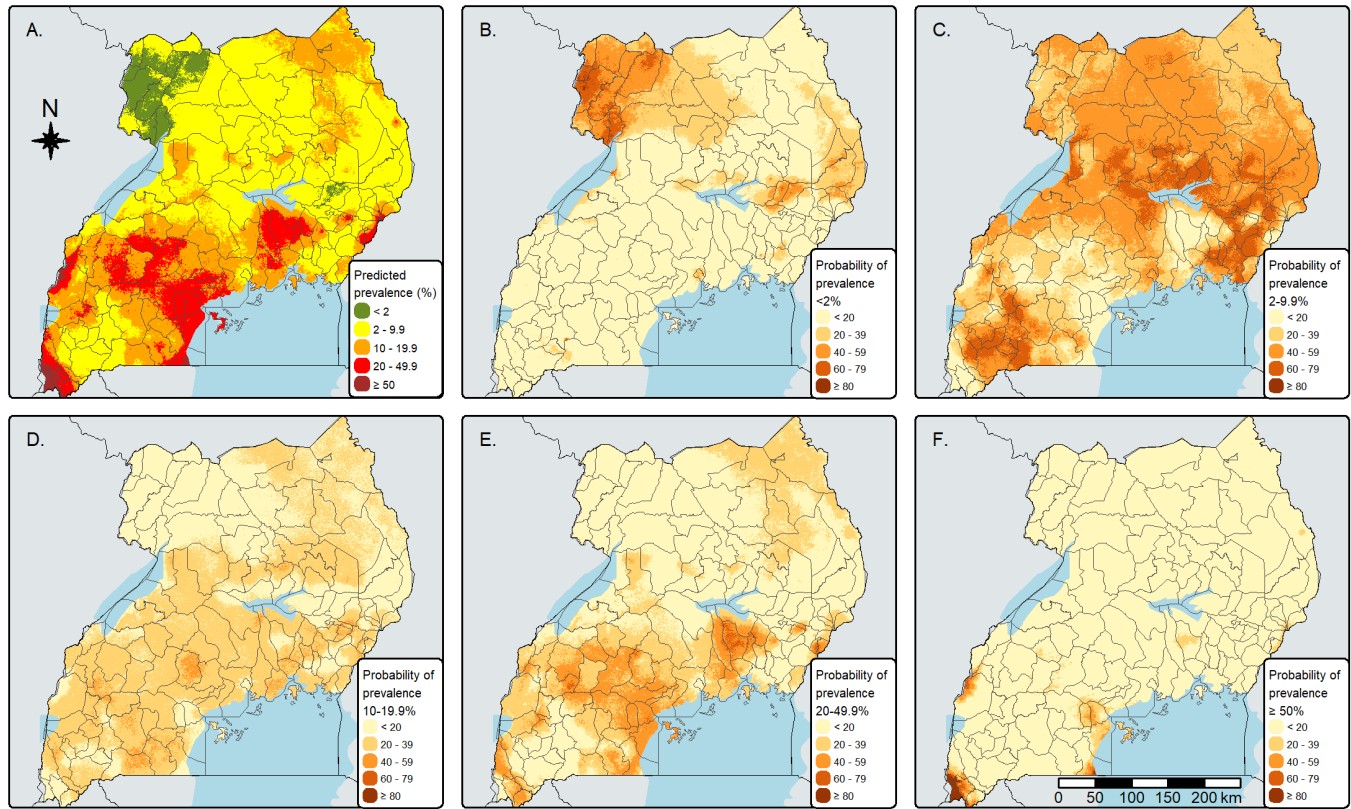

**Fig 2. Maps of median posterior predictive distribution of infection with any soil-transmitted helminth species (A) and the posterior probabilities of very low (B), low (C), moderate (D), high (E), and very high (F) prevalence of any soil-transmitted helminth infection among school-age children.** These maps were created using the tmap-package in R, and the basemap shapefiles were downloaded from ESPEN (2022). https://espen.afro.who.int/maps-data/data-query-tools/cartography-database [5] and https://data.humdata.org/dataset/cod-ab-uga [38].

**Table 3. Number of districts and estimated deworming tablet needs by endemicity category in Uganda.**

| Endemicity category | Prevalence category | WHO-recommended PC frequency | Districts (N) | SAC (N) | Tablets needed annually (N) |
|---|---|---|---|---|---|
| Very low | <2% | Event-based* | 2 | 175,029 | 0 |
| Low | 2–9.9% | 1 × every two years | 61 | 5,400,990 | 2,700,495 |
| Moderate | 10–19.9% | 1 × every year | 34 | 3,211,528 | 3,211,528 |
| High/very high | ≥20% | 2 × every year | 49 | 5,503,711 | 11,007,422 |
| Total | | | 146 | 14,291,258 | 16,919,445 |

WHO, World Health Organization; PC, preventive chemotherapy; SAC, school-age children (aged 5–14 years).

*Endemicity categories classified according to the WHO recommendations (*https://www.who.int/publications/b/73248*).

The 2025 SAC population estimates were obtained from the Uganda Ministry of Health.*WHO recommendations suggest that while PC distribution targeting entire at-risk groups may be suspended, it can continue in appropriate event-based settings. This analysis assumes that PC was suspended.

5.0% (95% BCI: 0.8–11.8%) for *A. lumbricoides*, 3.5% (0.7–9.3%) for *T. trichiura*, and 7.2% (5.7–11.1%) for hookworm. Hookworm was the predominant species, with a heterogeneous geographic distribution observed both within and between districts. For *A. lumbricoides* and *T. trichiura*, the highest prevalences were estimated at border and lake districts, highlighting potential control gaps that may require special attention and cross-border control efforts.

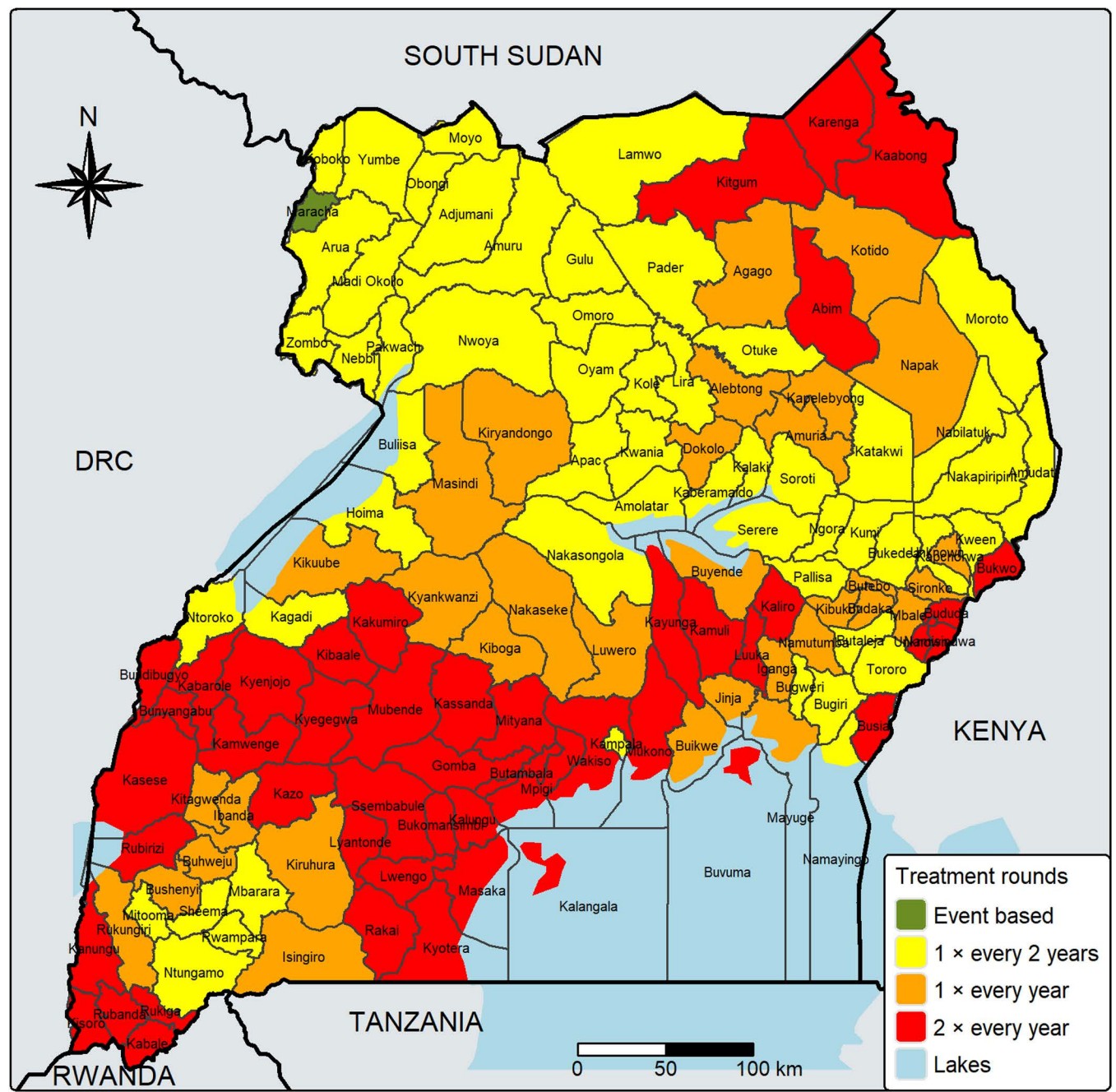

**Fig 3. Map of recommended soil-transmitted helminthiasis treatment rounds in Uganda by district based on results from predictive modeling.** This map was created using the tmap-package in R, and the basemap shapefiles were downloaded from ESPEN (2022) https://espen.afro.who.int/maps-data/data-query-tools/cartography-database [5] and. https://data.humdata.org/dataset/cod-ab-uga [38].

## Predictors

For *A. lumbricoides*, the minimum temperature in the coldest month and the proportion of sandy particles in the soil were important predictors. Other studies have reported contrasting associations, where an increase in minimum temperature was negatively [39] and positively associated with *A. lumbricoides* infection [40]. Previous studies have also reported conflicting findings regarding STH egg survival in sandy soils [41–43]. Additionally, although not statistically important in the final model, improved sanitation was an important predictor for *A. lumbricoides,* as it was included in 43 out of the 71 models assessed. Other studies have linked improved sanitation to lower *A. lumbricoides* prevalence [7]. Nightlights were negatively associated with hookworm infection. Nightlights are a good proxy indicator for SES and urbanization, as urban areas generally have more nightlights than rural areas. This finding aligns with previous studies showing that hookworm infection is more common in rural areas and less-developed parts of urban areas, where children are more likely to walk barefoot [44,45].

The finding that increased treatment was positively associated with hookworm infection was supported by the geographic distribution of treatment coverage (Fig B in S2 Appendix). This finding suggests that treatment coverage is higher in districts with high prevalence or insufficient information about the prevalence. Similar findings were observed in malaria, where areas with high malaria intervention coverage also had high malaria prevalence [46]. Low adherence to PC, treatment focusing mainly on children, and reinfection may also play a role. Given that hookworm reinfection is common and there is a potential for drug resistance to emerge, new control tools and combined therapies may need to be considered [47,48]. This finding could also be explained by poor sanitation, high population density, programmatic limitations such as suboptimal timing, or irregular PC campaigns that fail to break the STH transmission cycles. Conducting additional surveys in districts with sparse prevalence data and conducting behavioral and adherence studies, coupled with equity analyses, could help elucidate the reasons behind this unintuitive finding.

## Prevalence

BGM was performed at the species level to account for the distinct climatic and environmental drivers that influence the distribution and transmission dynamics of each STH species. We observed a relatively even distribution of hookworm prevalence across much of Uganda. However, the distribution of *A. lumbricoides* and *T. trichiura* was more focal, concentrated mainly in the border districts in the southwestern part of the country and areas close to the western shores of Lake Victoria. The high prevalence in these regions, particularly in Kisoro and Rubanda districts, has been previously reported [16]. The predicted surfaces help visualize pixel-level prevalence and highlight areas where immediate control efforts should be targeted.

Areas with very high endemicity of *T. trichiura* and *A. lumbricoides* may benefit from combination therapy, particularly albendazole and ivermectin. This approach may not only help reduce the burden in those specific locations but may also slow the development and spread of resistance to anthelmintic drugs [49]. Additionally, coordinated control efforts between countries may be required for sustainable STH control in the southwestern districts bordering the Democratic Republic of the Congo and Rwanda. Probability maps of endemicity levels can be used to identify potential hotspots where areas may not have shifted to a lower endemicity category after 5 years of treatment.

## Treatment frequency

We estimate that approximately 17.0 million tablets will be required for PC in 2025, representing a reduction of approximately 11 million tablets (~40%) compared to the amount required under universal, twice-yearly treatment for Uganda's 14.3 million SAC. If these WHO-recommended treatment frequencies were adopted based on our estimates, Uganda would be on track to meet WHO's target of halving the tablets required for PC by 2030 [34].

Given the high variability (uncertainty) of prevalence within several districts, we adopted a conservative approach to minimize under-treatment. Districts with high uncertainty were classified into the treatment category corresponding to the higher endemicity level. This led to several districts with posterior median prevalence estimates between 10% and 19.9% classified into the higher (≥20%) category. These districts should be considered for future surveys to increase available data, reduce estimation uncertainty, and enhance classification accuracy.

Furthermore, the variation in endemicity levels within districts suggests that a more accurate treatment classification could be achieved at the sub-district level. Tablet needs may be further reduced if the implementation unit is adjusted from the district to the sub-district level.

### Novelty of modeling approach

We employed a non-conventional variable selection strategy for inclusion in the final model. Prior studies have typically selected variables based primarily on p-values and significance levels, often overlooking the spatial correlation present in the data. Moreover, some studies suggest that statistically important variables are not necessarily strong predictors [50–52]. Since the current study's goal was to accurately estimate the prevalence and tablet needs for PC aimed at STH, selecting a model with the best predictive ability was crucial. We assessed the importance of variables included in the top similar models (ΔWAIC ≤4) with the least WAIC, then selected the model with the lowest RMSE as the best model for predicting prevalence in unsampled locations (15% of randomly selected locations). Additionally, we proposed an approach for assigning districts into treatment categories, taking into account uncertainty in the prevalence estimates. While the latter is entirely novel, the use of WAIC and cross-validation, rather than relying solely on statistically important predictors, has been recommended in other contexts before [31,50,53].

### Limitations

This study comes with several limitations. First, it integrates data from both school-based and community-based surveys, without accounting for school attendance status, as this information was not available in the community-based data. School-based surveys may underestimate STH prevalence, as they exclude out-of-school children, who are often at higher risk of infection. Second, prevalence estimates were based on a single stool sample with duplicate Kato-Katz thick smears, which is known to have lower sensitivity in low-prevalence and low-intensity settings compared to areas with higher STH burden [54]. Hence, the observed prevalence may underestimate the true prevalence, potentially leading to misclassification of treatment categories. Third, no surveys were conducted in the northeastern region of Uganda. This implies that the predicted prevalence estimates in this area should be interpreted with caution, as they were generated outside the domain of observed data. Future studies should survey this area, which has a higher prevalence of open defecation compared to the rest of the country. Additionally, the prediction uncertainty of the soil- and WASH-related predictors were not taken into account in our models, potentially affecting the prevalence prediction accuracy. Lastly, to avoid underestimating tablet needs, we took a conservative approach where districts with high uncertainty were assigned to a higher treatment frequency class. This may lead to over-estimation of tablet needs in those districts. Additional surveys are required to improve prevalence estimation and treatment classification.

### Conclusions

Using BGM, we harnessed environmental covariates and field survey data to classify districts based on WHO treatment recommendations. This resource-efficient approach generates critical evidence that can help control programs allocate treatment more effectively to those who need it most.

While universal biannual PC distribution may have been a necessary control strategy when prevalence was high across Uganda, our findings indicate that prevalence is now heterogeneous across the country, and as a result, treatment

frequency (and tablet needs) may be reduced in some areas. Epidemiologically-informed targeted PC should be considered to optimize the use of donated medicine, conserve human resources, and minimize unnecessary treatment of children who are no longer at high risk of infection.

## Supporting information

**S1 Appendix.** **Table A**. Description, sources, time period, and the spatial and temporal resolution of covariates. **Table B**. Geostatistical bivariate analyses with and without non-spatial random effect to determine variables associated with *A. lumbricoides* infection in Uganda. **Table C**. Geostatistical bivariate analyses with and without non-spatial random effect to determine variables associated with *T. trichiura* infection in Uganda. **Table D**. Geostatistical bivariate analyses with and without non-spatial random effect to determine variables associated with hookworm infection in Uganda. **Table E**. Posterior median and 95% Bayesian credible interval (BCI) estimates of parameters from the final multivariable geostatistical models by STH species. **Table F**. The prevalence, probability and treatment needs by district.
(XLSX)

**S2 Appendix.** **Fig A**. These maps display the observed and predicted values of the proportion of households with improved sanitation (A and D), improved drinking water (B and E) and open defecation (C and F). These maps were created using the tmap-package in R and the shapefiles were downloaded from https://data.humdata.org/dataset/cod-ab-uga. **Fig B**. The geographical distribution of hookworm prevalence by treatment coverage by district. This map was created using the tmap-package in R, and the shapefile was downloaded from ESPEN (2022) - https://espen.afro.who.int/maps-data/data-query-tools/cartography-database
(DOCX)

**S3 Appendix.** **Bayesian model formulation.**
(DOCX)

## Acknowledgments

We are grateful to all the Uganda Ministry of Health officers and district health teams who planned and conducted the surveys.

## Author contributions

**Conceptualization:** Bryan O. Nyawanda, Kristin M. Sullivan, Paul M. Emerson, Penelope Vounatsou.

**Data curation:** Bryan O. Nyawanda, Kristin M. Sullivan, Benjamin Tinkitina, Penelope Vounatsou.

**Formal analysis:** Bryan O. Nyawanda.

**Funding acquisition:** Kristin M. Sullivan, Paul M. Emerson, Penelope Vounatsou.

**Methodology:** Bryan O. Nyawanda, Penelope Vounatsou.

**Software:** Bryan O. Nyawanda, Penelope Vounatsou.

**Validation:** Bryan O. Nyawanda.

**Visualization:** Bryan O. Nyawanda.

**Writing – original draft:** Bryan O. Nyawanda.

**Writing – review & editing:** Bryan O. Nyawanda, Kristin M. Sullivan, Benjamin Tinkitina, Prudence Beinamaryo, Betty Nabatte, Hilda Kyarisiima, Alfred Mubangizi, Paul M. Emerson, Jürg Utzinger, Penelope Vounatsou.

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
