## [Decision Letter · Decision Letter 0]

11 May 2025

Geostatistical analysis to guide treatment decisions for soil-transmitted helminthiasis control in Uganda

Dear Dr. Vounatsou,

Thank you for submitting your manuscript to PLOS Neglected Tropical Diseases. After careful consideration, we feel that it has merit but does not fully meet PLOS Neglected Tropical Diseases's publication criteria as it currently stands. Therefore, we invite you to submit a revised version of the manuscript that addresses the points raised during the review process.

Please submit your revised manuscript within 60 days Jul 10 2025 11:59PM. If you will need more time than this to complete your revisions, please reply to this message or contact the journal office at plosntds@plos.org. Please include the following items when submitting your revised manuscript:

We look forward to receiving your revised manuscript.

Kind regards,

David J. Diemert, M.D.

Academic Editor

Eva Clark

Section Editor

Shaden Kamhawi

co-Editor-in-Chief

Paul Brindley

co-Editor-in-Chief

**Journal Requirements:**

At this stage, the following Authors/Authors require contributions: Bryan O. Nyawanda, Kristin M. Sullivan, Benjamin Tinkitina, Prudence Beinamaryo, Betty Nabatte, Hilda Kyarisiima, Alfred Mubangizi, Paul M. Emerson, Jürg Utzinger, and Penelope Vounatsou. Please ensure that the full contributions of each author are acknowledged in the "Add/Edit/Remove Authors" section of our submission form.

3) Some material included in your submission may be copyrighted. According to PLOSu2019s copyright policy, authors who use figures or other material (e.g., graphics, clipart, maps) from another author or copyright holder must demonstrate or obtain permission to publish this material under the Creative Commons Attribution 4.0 International (CC BY 4.0) License used by PLOS journals. Please closely review the details of PLOSu2019s copyright requirements here: PLOS Licenses and Copyright. If you need to request permissions from a copyright holder, you may use PLOS's Copyright Content Permission form.

Potential Copyright Issues:

i) Figures 1, 2, 3, and S2. Please (a) provide a direct link to the base layer of the map (i.e., the country or region border shape) and ensure this is also included in the figure legend; and (b) provide a link to the terms of use / license information for the base layer image or shapefile. We cannot publish proprietary or copyrighted maps (e.g. Google Maps, Mapquest) and the terms of use for your map base layer must be compatible with our CC BY 4.0 license.

4) Please amend your detailed Financial Disclosure statement. This is published with the article. It must therefore be completed in full sentences and contain the exact wording you wish to be published. Please ensure that the funders and grant numbers match between the Financial Disclosure field and the Funding Information tab in your submission form. Note that the funders must be provided in the same order in both places as well. State the initials, alongside each funding source, of each author to receive each grant. For example: "This work was supported by the National Institutes of Health (####### to AM; ###### to CJ) and the National Science Foundation (###### to AM).".

**Reviewers' Comments:**

Reviewer's Responses to Questions

**Key Review Criteria Required for Acceptance?**

**Methods**

-Are the objectives of the study clearly articulated with a clear testable hypothesis stated?

-Is the study design appropriate to address the stated objectives?

-Is the population clearly described and appropriate for the hypothesis being tested?

-Is the sample size sufficient to ensure adequate power to address the hypothesis being tested?

-Were correct statistical analysis used to support conclusions?

-Are there concerns about ethical or regulatory requirements being met?

Reviewer #1: The study design and statistical methods are largely appropriate to address the objectives of the analysis and there are not concerns about ethical or regulatory requirements. Some additional comments for consideration:

1. Page 7, line 131 - it would be good to know how many studies incorporated WASH and treatment data, and whether this was individual-level data or aggregate data.

2. Page 9, lines 181-184 - please clarify why the authors approach overall STH infection in this manner (equation based on individual STH species) as opposed to using empiric overall STH infection data (similar to how species-specific data is used). This analysis is focused on programmatic decision making, which

3. Page 9, lines 187-191 - please include any precedence of references for this approach.

4. Page 9, line 197 - what about treatment frequency for STH prevalence >50%?

5. Page 9, lines 202-204 - could not this mapping be the same as page 9 lines 184-187?

6. Page 10, line 216 - please specify what treatment categories 1-4 mean.

7. Please include the methods for producing the uncertainty estimates and maps.

Reviewer #2: (No Response)

Reviewer #3: Yes, and no concerns about ethical or regulatory requirements

**Results**

-Does the analysis presented match the analysis plan?

-Are the results clearly and completely presented?

-Are the figures (Tables, Images) of sufficient quality for clarity?

Reviewer #1: Overall the results are well presented and the figures are of sufficient quality. Some additional comments for consideration:

1. Page 10, line 228 - can the authors include a descriptive estimate for any STH infection?

2. Table 1 - it may be good for readers to understand why the rows do not necessarily add up to the overall models.

3. Table 1 - the authors could provide some insights as to their thoughts on why there is such a discrepancy in treatment as a predictor across the species.

4 Page 13 - could a model be developed for overall STH infection (in keeping with comment 2 for methods)?

5. Page 17, lines 319-321 - please clarify how these national predicted estimates were derived (e.g., in methods).

Reviewer #2: (No Response)

Reviewer #3: Yes

**Conclusions**

-Are the conclusions supported by the data presented?

-Are the limitations of analysis clearly described?

-Do the authors discuss how these data can be helpful to advance our understanding of the topic under study?

-Is public health relevance addressed?

Reviewer #1: The conclusions are supported and limitations are clearly described. Additional things to consider:

1. It would be good to have more understanding of the current Ugandan STH control program, this can be in the introduction. Drug requirement is only brought to attention in the discussion. Given a major focus is on how this analysis can optimise drug procurement the manuscript would benefit by providing more background to the Ugandan STH control program in the introduction.

2. Page 21, lines 409-412 - the authors could elaborate more on how geostatistical modelling can support drug procurement at smaller spatial scales.

Reviewer #2: (No Response)

Reviewer #3: Yes

**Editorial and Data Presentation Modifications?**

Reviewer #1: (No Response)

Reviewer #2: (No Response)

Reviewer #3: (No Response)

**Summary and General Comments**

Reviewer #1: The authors should be commended on a well thought-out and executed analysis. Their application of geostatistical modelling to guide programmatic decision making is informative. Please find in the relevant sections comments for consideration.

Reviewer #2: General

This paper adds to the evidence that geostatistical methods can enable more focused application of preventive chemotherapy, thus reducing the total cost of an effective control strategy. I have concerns about some aspects of the methodology and/or its exposition that I think need to be addressed before publication.

Specific

1. The authors acknowledge that their approach to model selection is “non-conventional.” I agree, which makes it all the more important that it be described clearly to allow others to emulate it. I found Table 1 hard to understand. The test states that “The relative frequency with which each predictor was included in the top models with similar WAIC is summarized in Table 1.” I therefore expected each set of three integers in each row of Table 1 to add to the number of models under consideration, which is clearly not the case. What am I missing?

2. I am sceptical of completely automated variable selection methods. In my opinion, contextual knowledge should also play a role, to distinguish between covariates for which there is or is not an a priori expectation of their importance. The authors should at least comment, in the discussion section, on how their preferred models do or do not have face validity in this sense.

3. I assume that the authors used the sample functionality within the INLA software, as this is essential for the validity of area-level probability statements. If so, this should be stated explicitly.

4. The assumption that the three specific prevalences are independent is dubious. What evidence (empirical or otherwise) is there to support this? How different would the results be if the “any STH” response was defined empirically.

5. The counter-intuitive finding that “increased treatment was positively significantly associated with STH infection” suggests strongly that the sampled locations were biased in favour of high-risk areas. The authors should describe what sampling designs (if any) were used and what are the implications for their findings. Incidentally, the use of “significantly” conflicts with their earlier statement about avoiding the use of this word having no place in Bayesian inference.

6. I could find no mention of prior distributions - these should be stated and justified. Also, for the preferred model, priors should be compared with posteriors to check that the inferences are not being driven by the priors.

7. Nor could I find any diagnostic checks on the preferred model.

8. An additional reference to the use of geostatistical models to enable more focused application of PC against STH is: Ediriweera et al (2019). Reassessment of the prevalence of soil-transmitted helminth infections in Sri Lanka to enable a more focused control programme: a cross-sectional national school survey with spatial modelling. Lancet Global Health, http://dx.doi.org/10.1016/S2214-109X(19)30253-0.

Reviewer #3: Authors provide a detailed and thoughtful use of geostatistics to identify optimal predictors of different STH species, and classiy sub-national areas of Uganda into policy relveant prevalnce thresholds. The article is well written, albeit perhaps a little too technical for the general readership of PLoS NTDs. The analysis is well performed and results are presented clearly . I have a few minor comments/clarifications which may strengthen the manuscript:

- In the intro, it might be good to mention that geostatistics is one of the recommended methodologies in the new WHO STH M&E guidelines. It has also been used in a couple of other settings to achieve similar goals (Kenya, Ethiopia), which authors could mention.

- The species specific point estimates presented in the abstract, the total of the 3 species is quite far form the any STH. I imagine this is due to seperate optimal covariate models being fit for each, but it will be confusing for non-technical audiences. I suggest providing an explanation in the body of the text, and consider removing these figures from the abstract - perhaps describing the overall species specific, and any STH, probability bands instead?

- The results have the requiste pills needed ina a given year to demonstrate the efficiency of sub-national targeting, but WHO guidelines work on a 2-year basis, could authors correct these calculations to show the 2 year tield.

- The analysis utilised data from 2 survey types: community and school. Was there any attempt to account for the likely variance between these two data sources (Community likely incorproate out of school children which means school based estimates may be artifically low).

- The validatoin of the different species specific models used an 85%/15% split of training vs. test data. Was there any basis for using this split?

- Table 1, the total number of models tested appears different for each species. Why was this?

- The approach of using a 10% level of probability seems novel, could the authors expand on this, namely why 10% was chosen. Also perhaps a couple of examples in text, for example, what would happen if probabilities were v_low 35%, low 35%, moderate 30%, high 0% or something like that. Though, perhaps I'm not understanding he classification algorithm correctly.

- It seems this article is jointly aiming to be a methods article, and also a policy discussion on updated MDA needs but also the use of geostats to improve efficiency. Could the authors add a bit more explanation on the latter, perhaps incorporate the table in the supplementary material which highlights the classification probabilities into the main text and spend a few sentences describing how this might be interpreted?

PLOS authors have the option to publish the peer review history of their article (what does this mean? ). If published, this will include your full peer review and any attached files.

**Do you want your identity to be public for this peer review?** For information about this choice, including consent withdrawal, please see our Privacy Policy .

Reviewer #1: No

Reviewer #2: No

Reviewer #3: No

**Figure resubmission:**

**Reproducibility:**



---

## [Decision Letter · Decision Letter 1]

11 Aug 2025

Dear Dr. Vounatsou,

We are pleased to inform you that your manuscript 'Geostatistical analysis to guide treatment decisions for soil-transmitted helminthiasis control in Uganda' has been provisionally accepted for publication in PLOS Neglected Tropical Diseases.

Best regards,

David J. Diemert, M.D.

Academic Editor

Eva Clark

Section Editor

Shaden Kamhawi

co-Editor-in-Chief

Paul Brindley

co-Editor-in-Chief

Reviewer's Responses to Questions

**Key Review Criteria Required for Acceptance?**

**Methods**

-Are the objectives of the study clearly articulated with a clear testable hypothesis stated?

-Is the study design appropriate to address the stated objectives?

-Is the population clearly described and appropriate for the hypothesis being tested?

-Is the sample size sufficient to ensure adequate power to address the hypothesis being tested?

-Were correct statistical analysis used to support conclusions?

-Are there concerns about ethical or regulatory requirements being met?

Reviewer #1: The authors have adequately attended to the peer review and the methods are suitable.

Reviewer #2: The authors have responded constructuvely to my comments on the original submission. In particular. they now either discuss, or include results relating to, my specific concerns about aspects of their methodology, allowing readers to make their own judgements on the robustness of their findings.

Reviewer #3: All comments in last review have been adequately addressed

**Results**

-Does the analysis presented match the analysis plan?

-Are the results clearly and completely presented?

-Are the figures (Tables, Images) of sufficient quality for clarity?

Reviewer #1: The authors have adequately attended to the peer review and the results are presented well.

Reviewer #2: Yes

Reviewer #3: All comments in last review have been adequately addressed

**Conclusions**

-Are the conclusions supported by the data presented?

-Are the limitations of analysis clearly described?

-Do the authors discuss how these data can be helpful to advance our understanding of the topic under study?

-Is public health relevance addressed?

Reviewer #1: The authors have adequately attended to the peer review and the discussion is well constructed.

Reviewer #2: Yes

Reviewer #3: All comments in last review have been adequately addressed

**Editorial and Data Presentation Modifications?**

Reviewer #1: (No Response)

Reviewer #2: I am happy to recommend acceptance of this version.

Reviewer #3: All comments in last review have been adequately addressed

**Summary and General Comments**

Reviewer #1: The authors should be commended on how well they have attended to the peer review.

Reviewer #2: A nice addition to the growing literature on the use of geospatial statistical methods in NTD research.

Reviewer #3: All comments in last review have been adequately addressed

PLOS authors have the option to publish the peer review history of their article (what does this mean? ). If published, this will include your full peer review and any attached files.

**Do you want your identity to be public for this peer review?** For information about this choice, including consent withdrawal, please see our Privacy Policy .

Reviewer #1: No

Reviewer #2: No

Reviewer #3: No